# Graph Structured Prediction Energy Networks

**Colin Graber**
cgraber2@illinois.edu

**Alexander Schwing**
aschwing@illinois.edu

Department of Computer Science
University of Illinois at Urbana-Champaign
Champaign, IL

## Abstract

For joint inference over multiple variables, a variety of structured prediction techniques have been developed to model correlations among variables and thereby improve predictions. However, many classical approaches suffer from one of two primary drawbacks: they either lack the ability to model high-order correlations among variables while maintaining computationally tractable inference, or they do not allow to explicitly model known correlations. To address this shortcoming, we introduce 'Graph Structured Prediction Energy Networks,' for which we develop inference techniques that allow to both model explicit local and implicit higher-order correlations while maintaining tractability of inference. We apply the proposed method to tasks from the natural language processing and computer vision domain and demonstrate its general utility.

## 1 Introduction

Many machine learning tasks involve joint prediction of a set of variables. For instance, semantic image segmentation infers the class label for every pixel in an image. To address joint prediction, it is common to use deep nets which model probability distributions independently over the variables (*e.g.*, the pixels). The downside: correlations between different variables aren't modeled explicitly.

A number of techniques, such as Structured SVMs [1], Max-Margin Markov Nets [2] and Deep Structured Models [3, 4], directly model relations between output variables. However, modeling the correlations between a large number of variables is computationally expensive and therefore generally impractical. As an attempt to address some of the fallbacks of classical high-order structured prediction techniques, Structured Prediction Energy Networks (SPENs) were introduced [5, 6]. SPENs assign a score to an entire prediction, which allows them to harness global structure. Additionally, because these models do not represent structure explicitly, complex relations between variables can be learned while maintaining tractability of inference. However, SPENs have their own set of downsides: Belanger and McCallum [5] mention, and we can confirm, that it is easy to overfit SPENs to the training data. Additionally, the inference techniques developed for SPENs do not enforce structural constraints among output variables, hence they cannot support structured scores and discrete losses. An attempt to combine locally structured scores with joint prediction was introduced very recently by Graber et al. [7]. However, Graber et al. [7] require the score function to take a specific, restricted form, and inference is formulated as a difficult-to-solve saddle-point optimization problem.

To address these concerns, we develop a new model which we refer to as 'Graph Structured Prediction Energy Network' (GSPEN). Specifically, GSPENs combine the capabilities of classical structured prediction models and SPENs and have the ability to explicitly model local structure when known or assumed, while providing the ability to learn an unknown or more global structure implicitly. Additionally, the proposed GSPEN formulation generalizes the approach by Graber et al. [7]. Concretely, inference in GSPENs is a maximization of a generally non-concave function w.r.t. structural constraints, for which we develop two inference algorithms.

We show the utility of GSPENs by comparing to related techniques on several tasks: optical character recognition, image tagging, multilabel classification, and named entity recognition. In general, we show that GSPENs are able to outperform other models. Our implementation is available at `https://github.com/cgraber/GSPEN`.

## 2  Background

Let $x \in \mathcal{X}$ represent the input provided to a model, such as a sentence or an image. In this work, we consider tasks where the outputs take the form $y = (y_1, \ldots, y_K) \in \mathcal{Y} := \prod_{k=1}^{K} \mathcal{Y}_k$, *i.e.*, they are vectors where the $k$-th variable's domain is the discrete and finite set $\mathcal{Y}_k = \{1, \ldots, |\mathcal{Y}_k|\}$. In general, the number of variables $K$ which are part of the configuration $y$ can depend on the observation $x$. However, for readability only, we assume all $y \in \mathcal{Y}$ contain $K$ entries, *i.e.*, we drop the dependence of the output space $\mathcal{Y}$ on input $x$.

All models we consider consist of a function $F(x, y; w)$, which assigns a score to a given configuration $y$ conditioned on input $x$ and is parameterized by weights $w$. Provided an input $x$, the inference problem requires finding the configuration $\hat{y}$ that maximizes this score, *i.e.*, $\hat{y} := \arg\max_{y \in \mathcal{Y}} F(x, y; w)$.

To find the parameters $w$ of the function $F(x, y; w)$, it is common to use a Structured Support Vector Machine (SSVM) (a.k.a. Max-Margin Markov Network) objective [1, 2]: given a multiset $\left\{ (x^i, y^i)_{i=1}^{N} \right\}$ of data points $(x^i, y^i)$ comprised of an input $x^i$ and the corresponding ground-truth configuration $y^i$, a SSVM attempts to find weights $w$ which maximize the margin between the scores assigned to the ground-truth configuration $y^i$ and the inference prediction:

$$\min_{w} \sum_{(x^i, y^i)} \max_{\hat{y} \in \mathcal{Y}} \left\{ F\left(x^i, \hat{y}; w\right) + L\left(\hat{y}, y^i\right) \right\} - F\left(x^i, y^i; w\right). \tag{1}$$

Hereby, $L\left(\hat{y}, y^i\right)$ is a task-specific and often discrete loss, such as the Hamming loss, which steers the model towards learning a margin between correct and incorrect outputs. Due to addition of the task-specific loss $L(\hat{y}, y^i)$ to the model score $F\left(x^i, \hat{y}; w\right)$, we often refer to the maximization task within Eq. (1) as loss-augmented inference. The procedure to solve loss-augmented inference depends on the considered model, which we discuss next.

**Unstructured Models.** Unstructured models, such as feed-forward deep nets, assign a score to each label of variable $y_k$ which is part of the configuration $y$, irrespective of the label choice of other variables. Hence, the final score function $F$ is the sum of $K$ individual scores $f_k(x, y_k; w)$, one for each variable:

$$F(x, y; w) := \sum_{k=1}^{K} f_k(x, y_k; w). \tag{2}$$

Because the scores for each output variable do not depend on the scores assigned to other output variables, the inference assignment is determined efficiently by independently finding the maximum score for each variable $y_k$. The same is true for loss-augmented inference, assuming that the loss decomposes into a sum of independent terms as well.

**Classical Structured Models.** Classical structured models incorporate dependencies between variables by considering functions that take more than one output space variable $y_k$ as input, *i.e.*, each function depends on a subset $r \subseteq \{1, \ldots, K\}$ of the output variables. We refer to the subset of variables via $y_r = (y_k)_{k \in r}$ and use $f_r$ to denote the corresponding function. The overall score for a configuration $y$ is a sum of these functions, *i.e.*,

$$F(x, y; w) := \sum_{r \in \mathcal{R}} f_r(x, y_r; w). \tag{3}$$

Hereby, $\mathcal{R}$ is a set containing all of the variable subsets which are required to compute $F$. The variable subset relations between functions $f_r$, *i.e.*, the structure, is often visualized using factor graphs or, generally, Hasse diagrams.

This formulation allows to explicitly model relations between variables, but it comes at the price of more complex inference which is NP-hard [8] in general. A number of approximations to this problem have been developed and utilized successfully (see Sec. 5 for more details), but the complexity of these methods scales with the size of the largest region $r$. For this reason, these models commonly consider only unary and pairwise regions, *i.e.*, regions with one or two variables.

Inference, *i.e.*, maximization of the score, is equivalent to the integer linear program

$$\max_{p \in \mathcal{M}} \sum_{r \in \mathcal{R}} \sum_{y_r \in \mathcal{Y}_r} p_r(y_r) f_r(x, y_r; w), \tag{4}$$

where each $p_r$ represents a marginal probability vector for region $r$ and $\mathcal{M}$ represents the set of $p_r$ whose marginal distributions are globally consistent, which is often called the marginal polytope. Adding an entropy term over the probabilities to the inference objective transforms the problem from maximum a-posteriori (MAP) to marginal inference, and pushes the predictions to be more uniform [9, 10]. When combined with the learning procedure specified above, this entropy provides learning with the additional interpretation of maximum likelihood estimation [9]. The training objective then also fits into the framework of Fenchel-Young Losses [11].

For computational reasons, it is common to relax the marginal polytope $\mathcal{M}$ to the local polytope $\mathcal{M}_L$, which is the set of all probability vectors that marginalize consistently for the factors present in the graph [9]. Since the resulting marginals are no longer globally consistent, *i.e.*, they are no longer guaranteed to arise from a single joint distribution, we write the predictions for each region using $b_r(y_r)$ instead of $p_r(y_r)$ and refer to them using the term "beliefs." Additionally, the entropy term is approximated using fractional entropies [12] such that it only depends on the factors in the graph, in which case it takes the form $H_{\mathcal{R}}(b) \coloneqq \sum_{r \in \mathcal{R}} \sum_{y_r \in \mathcal{Y}_r} -b_r(y_r) \log b_r(y_r)$.

**Structured Prediction Energy Networks.** Structured Prediction Energy Networks (SPENs) [5] were motivated by the desire to represent interactions between larger sets of output variables without incurring a high computational cost. The SPEN score function takes the following form:

$$F(x, p_1, \ldots, p_K; w) \coloneqq T\left(\bar{f}(x; w), p_1, \ldots, p_K; w\right), \tag{5}$$

where $\bar{f}(x; w)$ is a learned feature representation of the input $x$, each $p_k$ is a one-hot vector, and $T$ is a function that takes these two terms and assigns a score. This representation of the labels, *i.e.*, $p_k$, is used to facilitate gradient-based optimization during inference. More specifically, inference is formulated via the program:

$$\max_{p_k \in \Delta_k \forall k} T\left(\bar{f}(x; w), p_1, \ldots, p_K; w\right), \tag{6}$$

where each $p_k$ is constrained to lie in the $|\mathcal{Y}_k|$-dimensional probability simplex $\Delta_k$. This task can be solved using any constrained optimization method. However, for non-concave $T$ the inference solution might only be approximate.

**NLStruct.** SPENs do not support score functions that contain a structured component. In response, Graber et al. [7] introduced NLStruct, which combines a classical structured score function with a nonlinear transformation applied on top of it to produce a final score. Given a set $\mathcal{R}$ as defined previously, the NLStruct score function takes the following form:

$$F(x, p_{\mathcal{R}}; w) \coloneqq T\left(f_{\mathcal{R}}(x; w) \circ p_{\mathcal{R}}; w\right), \tag{7}$$

where $f_{\mathcal{R}}(x; w) \coloneqq (f_r(x, y_r; w))|_{r \in \mathcal{R}, y_r \in \mathcal{Y}_r}$ is a vectorized form of the score function for a classical structured model, $p_{\mathcal{R}} \coloneqq (p_r(y_r))|_{\forall r \in \mathcal{R}, \forall y_r \in \mathcal{Y}_r}$ is a vector containing all marginals, '$\circ$' is the Hadamard product, and $T$ is a scalar-valued function.

For this model, inference is formulated as a constrained optimization problem, where $\mathcal{Y}_{\mathcal{R}} \coloneqq \prod_{r \in \mathcal{R}} \mathcal{Y}_r$:

$$\max_{y \in \mathbb{R}^{|\mathcal{Y}_{\mathcal{R}}|}, p_{\mathcal{R}} \in \mathcal{M}} T(y; w) \text{ s.t. } y = f_{\mathcal{R}}(x; w) \circ p_{\mathcal{R}}. \tag{8}$$

Forming the Lagrangian of this program and rearranging leads to the saddle-point inference problem

$$\min_{\lambda} \max_{y} \left\{ T(y; w) - \lambda^T y \right\} + \max_{p_{\mathcal{R}} \in \mathcal{M}} \lambda^T \left( f_{\mathcal{R}}(x; w) \circ p_{\mathcal{R}} \right). \tag{9}$$

Notably, maximization over $p_{\mathcal{R}}$ is solved using techniques developed for classical structured models[1], and the saddle-point problem is optimized using the primal-dual algorithm of Chambolle and Pock [13], which alternates between updating $\lambda$, $y$, and $p_{\mathcal{R}}$.

| **Algorithm 1** Frank-Wolfe Inference for GSPEN | **Algorithm 2** Structured Entropic Mirror Descent Inference |
|---|---|
| 1: **Input:** Initial set of predictions $p_{\mathcal{R}}$; Input $x$; Factor graph $\mathcal{R}$ | 1: **Input:** Initial set of predictions $p_{\mathcal{R}}$; Input $x$; Factor graph $\mathcal{R}$ |
| 2: **for** $t = 1 \ldots T$ **do** | 2: **for** $t = 1 \ldots T$ **do** |
| 3: $\quad g \Leftarrow \nabla_{p_{\mathcal{R}}} F(x, p_{\mathcal{R}}; w)$ | 3: $\quad g \Leftarrow \nabla_{p_{\mathcal{R}}} F(x, p_{\mathcal{R}}; w)$ |
| 4: $\quad \widehat{p}_{\mathcal{R}} \Leftarrow \max_{\widehat{p}_{\mathcal{R}} \in \mathcal{M}_{\mathcal{R}}} \sum_{r \in \mathcal{R}, y_r \in \mathcal{Y}_r} \hat{p}_r(y_r) g_r(y_r)$ | 4: $\quad a \Leftarrow 1 + \ln p_{\mathcal{R}} + g / \sqrt{t}$ |
| 5: $\quad p_{\mathcal{R}} \Leftarrow p_{\mathcal{R}} + \frac{1}{t} (\widehat{p}_{\mathcal{R}} - p_{\mathcal{R}})$ | 5: $\quad p_{\mathcal{R}} \Leftarrow \max_{\hat{p}_{\mathcal{R}} \in \mathcal{M}} \sum_{r \in \mathcal{R}, y_r \in \mathcal{Y}_r} \hat{p}_r(y_r) a_r(y_r) + H_{\mathcal{R}}(\hat{p}_{\mathcal{R}})$ |
| 6: **end for** | 6: **end for** |
| 7: **Return:** $p_{\mathcal{R}}$ | 7: **Return:** $p_{\mathcal{R}}$ |

## 3 Graph Structured Prediction Energy Nets

Graph Structured Prediction Energy Networks (GSPENs) generalize all aforementioned models. They combine both a classical structured component as well as a SPEN-like component to score an entire set of predictions jointly. Additionally, the GSPEN score function is more general than that for NLStruct, and includes it as a special case. After describing the formulation of both the score function and the inference problem (Sec. 3.1), we discuss two approaches to solving inference (Sec. 3.2 and Sec. 3.3) that we found to work well in practice. Unlike the methods described previously for NLStruct, these approaches do not require solving a saddle-point optimization problem.

### 3.1 GSPEN Model

The GSPEN score function is written as follows:

$$F(x, p_{\mathcal{R}}; w) := T\left(\bar{f}(x; w), p_{\mathcal{R}}; w\right),$$

where vector $p_{\mathcal{R}} := (p_r(y_r))|_{r \in \mathcal{R}, y_r \in \mathcal{Y}_r}$ contains one marginal per region per assignment of values to that region. This formulation allows for the use of a structured score function while also allowing $T$ to score an entire prediction jointly. Hence, it is a combination of classical structured models and SPENs. For instance, we can construct a GSPEN model by summing a classical structured model and a multilayer perceptron that scores an entire label vector, in which case the score function takes the form $F(x, p_{\mathcal{R}}; w) := \sum_{r \in \mathcal{R}} \sum_{y_r \in \mathcal{Y}_r} p_r(y_r) f_r(x, y_r; w) + \text{MLP}(p_{\mathcal{R}}; w)$. Of course, this is one of many possible score functions that are supported by this formulation. Notably, we recover the NLStruct score function if we use $T(\bar{f}(x; w), p_{\mathcal{R}}; w) = T'(\bar{f}(x; w) \circ p_{\mathcal{R}}; w)$ and let $\bar{f}(x; w) = f_{\mathcal{R}}(x; w)$.

Given this model, the inference problem is

$$\max_{p_{\mathcal{R}} \in \mathcal{M}} T\left(\bar{f}(x; w), p_{\mathcal{R}}; w\right). \tag{10}$$

As for classical structured models, the probabilities are constrained to lie in the marginal polytope. In addition we also consider a fractional entropy term over the predictions, leading to

$$\max_{p_{\mathcal{R}} \in \mathcal{M}} T\left(\bar{f}(x; w), p_{\mathcal{R}}; w\right) + H_{\mathcal{R}}(p_{\mathcal{R}}). \tag{11}$$

In the classical setting, adding an entropy term relates to Fenchel duality [11]. However, the GSPEN inference objective does not take the correct form to use this reasoning. We instead view this entropy as a regularizer for the predictions: it pushes predictions towards a uniform distribution, smoothing the inference objective, which we empirically observed to improve convergence. The results discussed below indicate that adding entropy leads to better-performing models. Also note that it is possible to add a similar entropy term to the SPEN inference objective, which is mentioned by Belanger and McCallum [5] and Belanger et al. [6].

For inference in GSPEN, SPEN procedures cannot be used since they do not maintain the additional constraints imposed by the graphical model, *i.e.*, the marginal polytope $\mathcal{M}$. We also cannot use the inference procedure developed for NLStruct, as the GSPEN score function does not take the same form. Therefore, in the following, we describe two inference algorithms that optimize the program while maintaining structural constraints.

### 3.2 Frank-Wolfe Inference

The Frank-Wolfe algorithm [14] is suitable because the objectives in Eqs. (10, 11) are non-linear while the constraints are linear. Specifically, using [14], we compute a linear approximation of the

$$\min_{w} \sum_{\left(x^{(i)}, p_{\mathcal{R}}^{(i)}\right)} \left[ \max_{\hat{p}_{\mathcal{R}} \in \mathcal{M}} \left\{ T\left(\bar{f}(x;w), \hat{p}_{\mathcal{R}}; w\right) + L\left(\hat{p}_{\mathcal{R}}, p_{\mathcal{R}}^{(i)}\right) \right\} - T\left(\bar{f}\left(x^{(i)};w\right), p_{\mathcal{R}}^{(i)}; w\right) \right]_{+}$$

Figure 1: The GSPEN learning formulation, consisting of a Structured SVM (SSVM) objective with loss-augmented inference. Note that each $p_{\mathcal{R}}^{(i)}$ are one-hot representations of labels $y_i$.

objective at the current iterate, maximize this linear approximation subject to the constraints of the original problem, and take a step towards this maximum.

In Algorithm 1 we detail the steps to optimize Eq. (10). In every iteration we first calculate the gradient of the score function $F$ with respect to the marginals/beliefs using the current prediction as input. We denote this gradient using $g = \nabla_{p_{\mathcal{R}}} T\left(\bar{f}(x;w), p_{\mathcal{R}}; w\right)$. The gradient of $T$ depends on the specific function used and is computed via backpropagation. If entropy is part of the objective, an additional term of $-\ln(p_{\mathcal{R}}) - 1$ is added to this gradient.

Next we find the maximizing beliefs which is equivalent to inference for classical structured prediction: the constraint space is identical and the objective is a linear function of the marginals/beliefs. Hence, we solve this inner optimization using one of a number of techniques referenced in Sec. 5.

Convergence guarantees for Frank-Wolfe have been proven when the overall objective is concave, continuously differentiable, and has bounded curvature [15], which is the case when $T$ has these properties with respect to the marginals. This is true even when the inner optimization is only solved approximately, which is often the case due to standard approximations used for structured inference. When $T$ is non-concave, convergence can still be guaranteed, but only to a local optimum [16]. Note that entropy has unbounded curvature, therefore its inclusion in the objective precludes convergence guarantees. Other variants of the Frank-Wolfe algorithm exist which improve convergence in certain cases [17, 18]. We defer a study of these properties to future work.

### 3.3 Structured Entropic Mirror Descent

Mirror descent, another constrained optimization algorithm, is analogous to projected subgradient descent, albeit using a more general distance beyond the Euclidean one [19]. This algorithm has been used in the past to solve inference for SPENs, where entropy was used as the link function $\psi$ and by normalizing over each coordinate independently [5]. We similarly use entropy in our case. However, the additional constraints in form of the polytope $\mathcal{M}$ require special care.

We summarize the structured entropic mirror descent inference for the proposed model in Algorithm 2. Each iteration of mirror descent updates the current prediction $p_{\mathcal{R}}$ and dual vector $a$ in two steps: (1) $a$ is updated based on the current prediction $p_{\mathcal{R}}$. Using $\psi(p_{\mathcal{R}}) = -H_{\mathcal{R}}(p_{\mathcal{R}})$ as the link function, this update step takes the form $a = 1 + \ln p_{\mathcal{R}} + \frac{1}{\sqrt{t}}\left(\nabla_{p_{\mathcal{R}}} T\left(\bar{f}(x;w), p_{\mathcal{R}}; w\right)\right)$. As mentioned previously, the gradient of $T$ can be computed using backpropagation; (2) $p_{\mathcal{R}}$ is updated by computing the maximizing argument of the Fenchel conjugate of the link function $\psi^*$ evaluated at $a$. More specifically, $p_{\mathcal{R}}$ is updated via

$$p_{\mathcal{R}} = \max_{\hat{p}_{\mathcal{R}} \in \mathcal{M}} \sum_{r \in \mathcal{R}} \sum_{y_r \in \mathcal{Y}_r} \hat{p}_r(y_r) a_r(y_r) + H_{\mathcal{R}}(\hat{p}_{\mathcal{R}}), \qquad (12)$$

which is identical to classical structured prediction.

When the inference objective is concave and Lipschitz continuous (*i.e.*, when $T$ has these properties), this algorithm has also been proven to converge [19]. Unfortunately, we are not aware of any convergence results if the inner optimization problem is solved approximately and if the objective is not concave. In practice, though, we did not observe any convergence issues during experimentation.

### 3.4 Learning GSPEN Models

GSPENs assign a score to an input $x$ and a prediction $p$. An SSVM learning objective is applicable, which maximizes the margin between the scores assigned to the correct prediction and the inferred result. The full SSVM learning objective with added loss-augmented inference is summarized in Fig. 1. The learning procedure consists of computing the highest-scoring prediction using one of the inference procedures described in Sec. 3.2 and Sec. 3.3 for each example in a mini-batch and then updating the weights of the model towards making better predictions.

## 4 Experiments

To demonstrate the utility of our model and to compare inference and learning settings, we report results on the tasks of optical character recognition (OCR), image tagging, multilabel classification, and named entity recognition (NER). For each experiment, we use the following baselines: Unary is an unstructured model that does not explicitly model the correlations between output variables in any way. Struct is a classical deep structured model using neural network potentials. We follow the

| | Struct | SPEN | NLStruct | GSPEN |
|---|---|---|---|---|
| OCR (size 1000) | 0.40 s | 0.60 s | 68.56 s | 8.41 s |
| Tagging | 18.85 s | 30.49 s | 208.96 s | 171.65 s |
| Bibtex | 0.36 s | 11.75 s | – | 13.87 s |
| Bookmarks | 6.05 s | 94.44 s | – | 234.33 s |
| NER | 29.16 s | – | – | 99.83 s |

Table 1: Average time to compute inference objective and complete a weight update for one pass through the training data. We show all models trained for this work.

inference and learning formulation of [3], where inference consists of a message passing algorithm derived using block coordinate descent on a relaxation of the inference problem. SPEN and NLStruct represent the formulations discussed in Sec. 2. Finally, GSPEN represents Graph Structured Prediction Energy Networks, described in Sec. 3. For GSPENs, the inner structured inference problems are solved using the same algorithm as for Struct. To compare the run-time of these approaches, Table 1 gives the average epoch compute time (*i.e.*, time to compute the inference objective and update model weights) during training for our models for each task. In general, GSPEN training was more efficient with respect to time than NLStruct but, expectedly, more expensive than SPEN. Additional experimental details, including hyper-parameter settings, are provided in Appendix A.2.

### 4.1 Optical Character Recognition (OCR)

For the OCR experiments, we generate data by selecting a list of 50 common 5-letter English words, such as 'close,' 'other,' and 'world.' To create each data point, we choose a word from this list and render each letter as a 28x28 pixel image by selecting a random image of the letter from the Chars74k dataset [20], randomly shifting, scaling, rotating, and interpolating with a random background image patch. A different pool of backgrounds and letter images was used for the training, validation, and test splits of the

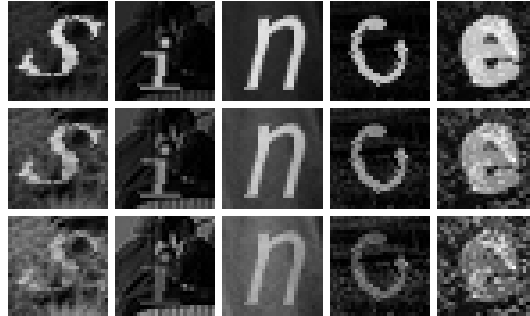

Figure 2: OCR sample data points with different interpolation factors $\alpha$.

data. The task is to identify the words given 5 ordered images. We create three versions of this dataset using different interpolation factors of $\alpha \in \{0.3, 0.5, 0.7\}$, where each pixel in the final image is computed as $\alpha x_{\text{background}} + (1 - \alpha)x_{\text{letter}}$. See Fig. 2 for a sample from each dataset. This process was deliberately designed to ensure that information about the structure of the problem (*i.e.*, which words exist in the data) is a strong signal, while the signal provided by each individual letter image can be adjusted. The training, validation, and test set sizes for each dataset are 10,000, 2,000, and 2,000, respectively. During training we vary the training data to be either 200, 1k or 10k.

To study the inference algorithm, we train four different GSPEN models on the dataset containing 1000 training points and using $\alpha = 0.5$. Each model uses either Frank-Wolfe or Mirror Descent and included/excluded the entropy term. To maintain tractability of inference, we fix a maximum iteration count for each model. We additionally investigate the effect of this maximum count on final performance. Additionally, we run this experiment by initializing from two different Struct models, one being trained using entropy during inference and one being trained without entropy. The results for this set of experiments are shown in Fig. 3a. Most configurations perform similarly across the number of iterations, indicating these choices are sufficient for convergence. When initializing from the models trained without entropy, we observe that both Frank-Wolfe without entropy and Mirror Descent with entropy performed comparably. When initializing from a model trained with entropy, the use of mirror descent with entropy led to much better results.

The results for all values of $\alpha$ using a train dataset size of 1000 are presented in Fig. 3b, and results for all train dataset sizes with $\alpha = 0.5$ are presented in Fig. 3c. We observe that, in all cases, GSPEN outperforms all baselines. The degree to which GSPEN outperforms other models depends most on the amount of train data: with a sufficiently large amount of data, SPEN and GSPEN perform comparably. However, when less data is provided, GSPEN performance does not drop as sharply as that of SPEN initially. It is also worth noting that GSPEN outperformed NLStruct by a large margin.

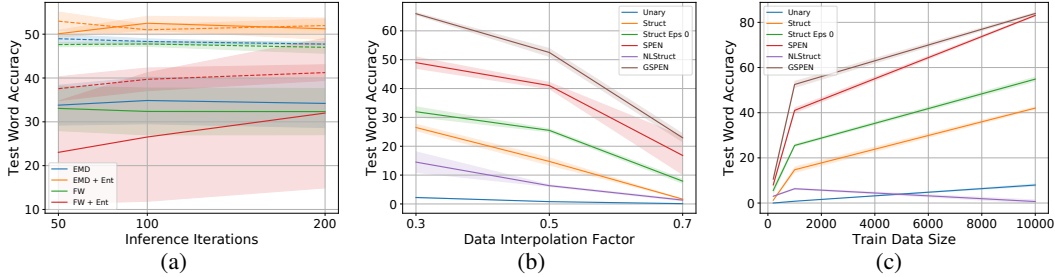

(a)            (b)            (c)

Figure 3: Experimental results on OCR data. The dashed lines in (a) represent models trained from Struct without entropy, while solid lines represent models trained from Struct with entropy.

The NLStruct model is less stable due to its saddle-point formulation. Therefore it is much harder to obtain good performance with this model.

## 4.2 Image Tagging

Next, we evaluate on the MIRFLICKR25k dataset [21], which consists of 25,000 images taken from Flickr. Each image is assigned a subset of 24 possible tags. The train/val/test sets for these experiments consist of 10,000/5,000/10,000 images, respectively.

We compare to NLStruct and SPEN. We initialize the structured portion of our GSPEN model using the pre-trained DeepStruct model described by Graber et al. [7], which consists of unary potentials produced from an AlexNet architecture [22] and linear pairwise potentials of the form $f_{i,j}(y_i, y_j, W) = W_{i,j,x_i,x_j}$, *i.e.*, containing one weight per pair in the graph per assignment of values to that pair. A fully-connected pairwise graph was used. The $T$ function for our GSPEN model consists of a 2-layer MLP with 130 hidden units. It takes as input a concatenation of the unary potentials generated by the AlexNet model and the current prediction. Additionally, we train a SPEN model with the same number of layers and hidden units. We used Frank-Wolfe without entropy for both SPEN and GSPEN inference.

The results are shown in Fig. 4. GSPEN obtains similar test performance to the NLStruct model, and both outperform SPEN. However, the NL-Struct model was run for 100 iterations during inference without reaching 'convergence' (change of objective smaller than threshold), while the GSPEN model required an average of 69 iterations to converge at training time and 52 iterations to converge at test time. Our approach has the advantage of requiring fewer variables to maintain during inference and requiring fewer iterations of inference to converge. The final test losses for SPEN, NLStruct and GSPEN are 2.158, 2.037, and 2.029, respectively.

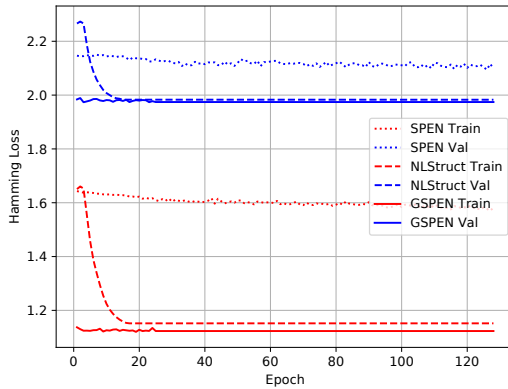

Figure 4: Results for image tagging.

## 4.3 Multilabel Classification

We use the Bibtex and Bookmarks multilabel datasets [23]. They consist of binary-valued input feature vectors, each of which is assigned some subset of 159/208 possible labels for Bibtex/Bookmarks, respectively. We train unary and SPEN models with architectures identical to [5] and [24] but add dropout layers. In addition, we further regularize the unary model by flipping each bit of the input vectors with probability 0.01 when sampling mini-batches. For Struct and GSPEN, we generate a graph by first finding the label variable that is active in most training data label vectors and add edges connecting every other variable to this most active one. Pairwise potentials are generated by passing the input vector through a 2-layer MLP with 1k hidden units. The GSPEN model is trained by starting from the SPEN model, fixing its parameters, and training the pairwise potentials.

The results are in Table 2 alongside those taken from [5] and [24]. We found the Unary models to perform similarly to or better than previous best results. Both SPEN and Struct are able to improve upon these Unary results. GSPEN outperforms all configurations, suggesting that the contributions of the SPEN component and the Struct component to the score function are complementary.

Table 2: Multilabel classification results for all models. All entries represent macro F1 scores. The top results are taken from the cited publications.

| | Bibtex | | Bookmarks | |
| | Validation | Test | Validation | Test |
|---|---|---|---|---|
| SPEN [5] | – | 42.2 | – | 34.4 |
| DVN [24] | – | 44.7 | – | 37.1 |
| Unary | 43.3 | 44.1 | 38.4 | 37.4 |
| Struct | 45.8 | 46.1 | 39.7 | 38.9 |
| SPEN | 46.6 | 46.5 | 40.2 | 39.2 |
| GSPEN | **47.5** | **48.6** | **41.2** | **40.7** |

Table 3: Named Entity Recognition results for all models. All entries represent F1 scores averaged over five trials.

| | Avg. Val. | Avg. Test |
|---|---|---|
| Struct [26] | $94.88 \pm 0.18$ | $91.37 \pm 0.04$ |
| + GSPEN | $\mathbf{94.97} \pm 0.16$ | $\mathbf{91.51} \pm 0.17$ |
| Struct [27] | $95.88 \pm 0.10$ | $\mathbf{92.79} \pm 0.08$ |
| + GSPEN | $\mathbf{95.96} \pm 0.08$ | $92.69 \pm 0.17$ |

## 4.4 NER

We also assess suitability for Named Entity Recognition (NER) using the English portion of the CoNLL 2003 shared task [25]. To demonstrate the applicability of GSPEN for this task, we transformed two separate models, specifically the ones presented by Ma and Hovy [26] and Akbik et al. [27], into GSPENs by taking their respective score functions and adding a component that jointly scores an entire set of predictions. In each case, we first train six instances of the structured model using different random initializations and drop the model that performs the worst on validation data. We then train the GSPEN model, initializing the structured component from these pre-trained models.

The final average performance is presented in Table 3, and individual trial information can be found in Table 4 in the appendix. When comparing to the model described by Ma and Hovy [26], GSPEN improves the final test performance in four out of the five trials, and GSPEN has a higher overall average performance across both validation and test data. Compared to Akbik et al. [27], on average GSPEN's validation score was higher, but it performed slightly worse at test time. Overall, these results demonstrate that it is straightforward to augment a task-specific structured model with an additional prediction scoring function which can lead to improved final task performance.

## 5 Related Work

A variety of techniques have been developed to model structure among output variables, originating from seminal works of [1, 2, 28]. These works focus on extending linear classification, both probabilistic and non-probabilistic, to model the correlation among output variables. Generally speaking, scores representing both predictions for individual output variables and for combinations of output variables are used. A plethora of techniques have been developed to solve inference for problems of this form, *e.g.*, [9, 12, 29–62]. As exact inference for general structures is NP-hard [8], early work focused on tractable exact inference. However, due to interest in modeling problems with intractable structure, a plethora of approaches have been studied for learning with approximate inference [63–70]. More recent work has also investigated the role of different types of prediction regularization, with Niculae et al. [10] replacing the often-used entropy regularization with an L2 norm and Blondel et al. [11] casting both as special cases of a Fenchel-Young loss framework.

To model both non-linearity and structure, deep learning and structured prediction techniques were combined. Initially, local, per-variable score functions were learned with deep nets and correlations among output variables were learned in a separate second stage [71, 72]. Later work simplified this process, learning both local score functions and variable correlations jointly [3, 4, 73–75].

Structured Prediction Energy Networks (SPENs), introduced by Belanger and McCallum [5], take a different approach to modeling structure. Instead of explicitly specifying a structure a-priori and enumerating scores for every assignment of labels to regions, SPENs learn a function which assigns a score to an input and a label. Inference uses gradient-based optimization to maximize the score w.r.t. the label. Belanger et al. [6] extend this technique by unrolling inference in a manner inspired by Domke [76]. Both approaches involve iterative inference procedures, which are slower than feed-forward prediction of deep nets. To improve inference speed, Tu and Gimpel [77] learn a neural net to produce the same output as the gradient-based methods. Deep Value Networks [24] follow the same approach of Belanger and McCallum [5] but use a different objective that encourages the score to equal the task loss of the prediction. All these approaches do not permit to include known structure. The proposed approach enables this.

Our approach is most similar to our earlier work [7], which combines explicitly-specified structured potentials with a SPEN-like score function. The score function of our earlier work is a special case of the one presented here. In fact, earlier we required a classical structured prediction model as an intermediate layer of the score function, while we don't make this assumption any longer. Additionally, in our earlier work we had to solve inference via a computationally challenging saddle-point objective. Another related approach is described by Vilnis et al. [78], whose score function is the sum of a classical structured score function and a (potentially non-convex) function of the marginal probability vector $p_{\mathcal{R}}$. This is also a special case of the score function presented here. Additionally, the inference algorithm they develop is based on regularized dual averaging [79] and takes advantage of the structure of their specific score function, *i.e.*, it is not directly applicable to our setting.

# 6    Conclusions

The developed GSPEN model combines the strengths of several prior approaches to solving structured prediction problems. It allows machine learning practitioners to include inductive bias in the form of known structure into a model while implicitly capturing higher-order correlations among output variables. The model formulation described here is more general than previous attempts to combine explicit local and implicit global structure modeling while not requiring inference to solve a saddle-point problem.

**Acknowledgments**

This work is supported in part by NSF under Grant No. 1718221 and MRI #1725729, UIUC, Samsung, 3M, Cisco Systems Inc. (Gift Award CG 1377144) and Adobe. We thank NVIDIA for providing GPUs used for this work and Cisco for access to the Arcetri cluster.

## Footnotes

[1] As mentioned, solving the maximization over $p_{\mathcal{R}}$ tractably might require relaxing the marginal polytope $\mathcal{M}$ to the local marginal polytope $\mathcal{M}_L$. For brevity, we will not repeat this fact whenever an inference problem of this form appears throughout the rest of this paper.

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
