[Supplementary Material · appendix.pdf]

Table 4: All Trials for NER experiments.

| Trial | Ma and Hovy [26] | | + GSPEN | |
| | Val. F1 | Test F1 | Val. F1 | Test F1 |
|---|---|---|---|---|
| 1 | 94.94 | 91.36 | 94.85 | 91.25 |
| 2 | 94.67 | 91.37 | 94.76 | 91.53 |
| 3 | 94.74 | 91.32 | 95.07 | 91.60 |
| 4 | 94.94 | 91.35 | 95.08 | 91.47 |
| 5 | 95.12 | 91.44 | 95.10 | 91.71 |
| | Akbik et al. [27] | | + GSPEN | |
| 1 | 0.9576 | 0.9284 | 0.9587 | 0.9288 |
| 2 | 0.9601 | 0.9277 | 0.9599 | 0.9246 |
| 3 | 0.9581 | 0.9271 | 0.9588 | 0.9271 |
| 4 | 0.9592 | 0.9291 | 0.9600 | 0.9280 |
| 5 | 0.9590 | 0.9272 | 0.9604 | 0.9260 |

# A   Appendix

In this Appendix, we first present additional experimental results for the NER before providing additional details on experiments, including model architectures and selection of hyper-parameters.

## A.1   Additional Experimental Results

Table 4 contains the results for all trials of the NER experiments. When comparing to Ma and Hovy [26], we outperform their model on both validation and test data in four out of five trials. When comparing to Akbik et al. [27], we outperform their model on validation data in four out of five trials, but only outperform their model on test data in one trial.

## A.2   Additional Experimental Details

**General Details:**   Unless otherwise specified, all Struct models were trained by using the corresponding pre-trained Unary model, fixing these parameters, and training pairwise potentials. All SPEN models were trained by using the pre-trained Unary model, fixing these parameters, and training the $T$ function. Early stopping based on task performance on validation was used to select the number of epochs for training. For SPEN, GSPEN, and Struct models, loss-augmented inference was used where the loss function equals the sum of the 0-1 losses per output variable, *i.e.*, $L(\hat{y}, y) \coloneqq \sum_{i=1}^{n} \mathbf{1}[\hat{y}_i \neq y_i]$ where $n$ is the number of output variables.

**OCR:**   The Unary model is a single 3-layer multilayer perceptron (MLP) with ReLU activations, hidden layer sizes of 200, and a dropout layer after the first linear layer with keep probability 0.5. Scores for each image were generated by independently passing them into this network. Both Struct and GSPEN use a graph with one pairwise region per pair of adjacent letters, for a total of 4 pairs. Linear potentials are used, containing one entry per pair per set of assignments of values to each pair. The score function for both SPEN and GSPEN takes the form $F(x, p; w) = \sum_{r \in \mathcal{R}} \sum_{y_r \in \mathcal{Y}_r} p_r(y_r) b_r(x, y; w) + T(B(x), p)$, where in the SPEN case $\mathcal{R}$ contains only unary regions and in the GSPEN case $\mathcal{R}$ consists of the graph used by Struct. Each $b_r$ represents the outputs of the same model as Unary/Struct for SPEN/GSPEN, respectively, and $B(x)$ represents the vector $(b_r(x, y_i; w)|_{y_r \in \mathcal{Y}_r})$. For every SPEN and GSPEN model trained, $T$ is a 2-layer MLP with softplus activations, an output size of 1, and either 500, 1000, or 2000 hidden units. These hidden sizes as well as the number of epochs of training for each model were determined based on task performance on the validation data. Message-passing inference used in both Struct and GSPEN ran for 10 iterations. GSPEN models were trained by using the pre-trained Struct model, fixing these parameters, and training the $T$ function. The NLStruct model consisted of a 2-layer MLP with 2834 hidden units, an output size of 1, and softplus activations. We use the same initialization described by Graber et al. [7] for their word recognition experiments, where the first linear layer was initialized to the identity matrix and the second linear layer was initialized to a vector of all 1s. NLStruct models were

initialized from the Struct models trained without entropy and used fixed potentials. The inference configuration described by Graber et al. [7] was used, where inference was run for 100 iterations with averaging applied over the final 50 iterations.

All settings for the OCR experiments used a mini-batch size of 128 and used the Adam optimizer, with Unary, SPEN, and GSPEN using a learning rate of $10^{-4}$ and Struct using a learning rate of $10^{-3}$. Gradients were clipped to a norm of 1 before updates were applied. Inference in both SPEN and GSPEN were run for a maximum of 100 iterations. Inference was terminated early for both models if the inference objective for all datapoints in the minibatch being processed changed by less than 0.0001.

Three different versions of every model, initialized using different random seeds, were trained for these experiments. The plots represent the average of these trials, and the error represented is the standard deviation of these trials.

**Tagging:**   The SPEN and GSPEN models for the image tagging experiment used the same scoring function form as for the OCR experiments. The $T$ model in both cases is a 2-layer MLP with softplus activations and 130 hidden units. Both GSPEN and SPEN use the same score function as in the OCR experiments, with the exception that the $T$ function used for GSPEN is only a function of the beliefs and does not include the potentials as input. Both models were trained using gradient descent with a learning rate of $10^{-2}$, a momentum of 0.9, and a mini-batch size of 128. Once again, only the $T$ component was trained for GSPEN, and the pairwise potentials were initialized to a Struct model trained using the settings described in Graber et al. [7].

The message-passing procedure used to solve the inner optimization problem for GSPEN was run for 100 iterations per iteration of Frank-Wolfe. Inference for SPEN and GSPEN was run for 100 iterations and was terminated early if the inference objective for all datapoints in the minibatch being processed changed by less than 0.0001.

**Multilabel Classification:**   For the Bibtex dataset, 25 percent of the training data was set aside to be used as validation data; this was not necessary for Bookmarks, which has a pre-specified validation dataset. For prediction in both datasets and for all models, a threshold determining the boundary between positive/negative label predictions was tuned on the validation dataset.

For the Bibtex dataset, the Unary model consists of a 3-layer MLP taking the binary feature vectors as input and returning a 159-dimensional vector representing the potentials for label assignments $y_i = 1$; the potentials for $y = 0$ are fixed to 0. The Unary model uses ReLU activations, hidden unit sizes of 150, and dropout layers before the first and second linear layers with keep probability of 0.5. The Struct model consists of a 2-layer MLP which also uses the feature vector as input, and it contains 1000 hidden units and ReLU activations. The SPEN model uses the same scoring function form as used in the previous experiments, except the $T$ function is only a function of the prediction vector and does not use the unary potentials as input. The $T$ model consists of a 2-layer MLP which takes the vector $(p_i(y_i = 1))_{i=1}^{59}$ as input. This model has 16 hidden units, an output size of 1, and uses softplus activations. The GSPEN model was trained by starting from the SPEN model, fixing these parameters, and training a pairwise model with the same architecture as the Struct model.

For the bookmarks dataset, the models use the same architectures with slightly different configurations. the Unary model consists of a similar 3-layer MLP, except dropout is only applied before the second linear layer. The Struct model uses the same architecture as the one trained on the Bibtex data. The $T$ model for SPEN/GSPEN uses 15 hidden units.

For both datasets and for both SPEN and GSPEN, mirror descent was used for inference with an additional entropy term with a coefficient of 0.1; for Struct, a coefficient of 1 was used. Inference was run for 100 iterations, with early termination as described previously using the same threshold. For Struct and GSPEN, message passing inference was run for 5 iterations. The Unary model was trained using gradient descent with a learning rate of $10^{-2}$ and a momentum of 0.9, while Struct, SPEN and GSPEN were trained using the Adam optimizer with a learning rate of $10^{-4}$.

**NER:**   Both structured model baselines were trained using code provided by the authors of the respective papers. In both cases, hyperparameter choices for these structured models were chosen to be identical to the the choices made from their original works. For completeness, we will review these choices.

The structured model of Ma and Hovy [26] first produces a vector for each word in the input sentence by concatenating two vectors: the first is a 100-dimensional embedding for the word, which is initialized from pre-trained GloVe embeddings [78] and fine-tuned. The second is the output of a 1-D convolutional deep net with 30 filters of length 3 taking as input 30-dimensional character embeddings for each character in the word. These representations are then passed into a 1-layer bidirectional LSTM with a hidden state size of 256, which is passed through a linear layer followed by an ELU activation to produce an intermediate representation. Unary/pairwise graphical model scores are finally obtained by passing this intermediate representation through two further linear layers. Predictions are made using the Viterbi algorithm. Dropout is applied to the embeddings before they are fed into the RNN (zeroing probability of $0.5511$, corresponding to two separate dropout layers with zeroing probability of $0.33$ being applied) and to the output hidden states of the RNN (zeroing probability of $0.5$). The GSPEN models in this setting were trained by initializing the structured component from the pre-trained models and fixing them – that is, only the parameters in the MLP were trained during this step. Due to the fact that the input sentences are of varying size, we zero-pad all inputs of the MLP to the maximum sequence length. Dropout with a zeroing probability of $0.75$ was additionally applied to the inputs of the MLP. Inference was conducted using mirror descent with added entropy and convergence threshold of $0.1$. For both the structured baseline and GSPEN, model parameters were trained for 200 epochs using SGD with initial learning rate of $0.01$, which was decayed every epoch using the formula $\text{lr}(\text{epoch}) = \frac{0.01}{1+\text{epoch}\cdot 0.05}$. The structured baseline was trained with a mini-batch size of 16, while the GSPEN model used a mini-batch size of 32 during training. A larger batch size was used for the GSPEN model to decrease the amount of time to complete one pass through the data.

Akbik et al. [27] use a concatenation of three different pre-trained embeddings per token as input to the bidirectional LSTM. The first is generated by a bidirectional LSTM which takes character-level embeddings as input and is pre-trained using a character-based language modeling objective (see Akbik et al. [27] for more details). The other two embeddings are GloVe word embeddings [78], and task-trained character-based embeddings (as specified by Lample et al. [79]). During training, these embeddings are fine-tuned by passing them through a linear layer whose parameters are learned. The embeddings are passed into a 1-layer bidirectional LSTM with a hidden state size of 256. Unary scores are generated from the outputs of the LSTM by passing them through a linear layer; pairwise scores consist of a matrix of scores for every pair of labels, which are shared across sentence indices. In this setting, the GSPEN models were trained by initializing the structured component from the pre-trained models and then fine-tuning all of the model parameters. Mirror descent with added entropy was used for GSPEN inference with a convergence threshold of $0.1$. For both the structured baseline and GSPEN, model parameters were trained for a maximum of 150 epochs using SGD with mini-batch size of 32 and initial learning rate of $0.1$, which was decayed by $0.5$ when the training loss did not decrease past its current minimum for 3 epochs. Training was terminated early if the learning rate fell below $10^{-4}$.