[Reviews · NeurIPS 2019]

Reviewer 1



This paper addresses weaknesses in structured prediction energy networks, which estimate a scalar score for a set of structured variables, by allowing multiple separate factors to be added together to produce such a score. The extension is non-trivial, requiring different inference algorithms to (locally) optimize the corresponding inference problem while satisfying constraints on marginals shared over factors. It also unifies existing work at the intersections of deep learning and structured prediction. The extensive experiments in the paper are a strong point. Care has been taken to provide good comparisons. The OCR datasets are constructed to show the benefits of modeling the structured output. Should the x-axis label of Fig 1a be “iterations” instead of training examples? The variability in test performance based on optimization algorithm in Fig 1a is somewhat alarming. Does inference optimization method choice become a “black art” in this approach? Despite this, the performance of GSPEN is at least very similar to (image tagging) or better than (other experiments) the comparison methods. Overall, this is an impressive paper that addresses key gaps in structured prediction for neural network/deep learning methods that reside between having explicit structural assumptions and allowing flexibility.

Reviewer 2



Based on structured SVM, the authors combine the structured prediction and the learning using hinge loss, the results is a novel model, Graph Structured Prediction Energy Networks. Overall the model is novel and the theory is mostly solid. However, I have some concerns about the inference part. 1. Marginal Polytope. The relaxation of the marginal polytope is always tricky for structured prediction. A loose relaxation might result in an efficient algorithm, but the bad quality of the solution. Tight relaxation is often unaffordable in terms of complexity. In the Inference algorithm, it is not clear what kind of relaxation of M is used. 2. Running time. The inference of a structured prediction model can always be slow. Can the authors provide a detailed running time of the proposed model on each application? ====== After Rebuttal ====== In the rebuttal, the authors still fail to give a detailed description of the local marginal polytope. It is true that GSPEN allows the practitioner to select a structured prediction algorithm (associated with some relaxation of M). However, in practice the choice of relaxation is tricky. The authors claim that they are using the local marginal polytope in [43]. However, there are also different versions of local marginal polytope in [43], e.g. Bethe approximation and Kikuchi approximation. It is also important that different approximation may result in different time complexity of inference algorithms. A tighter relaxation (e.g. Kikuchi approximation) may have a large complexity for each iteration but requires fewer iterations to reach a higher accuracy than a looser relaxation (e.g. Bethe approximation). Based on this, the complexity provided in the rebuttal might be too rough.

Reviewer 3



+ This paper is clearly motivated. The paper aims to address the limitation of structured output prediction methods, which struggle with explicitly modeling correlations between a large number of output variables. To this aim, the paper makes an incremental methodological contribution by extending a recent approach, Structured Prediction Energy Networks, by Belanger and McCallum (refs. [4,5]). + The paper demonstrably advances state of the art. Authors compare inference and learning settings and report results on the tasks of optical character recognition, image tagging, multilabel classification, and named entity recognition. - While empirical results support the claims of this paper, they are not supported by theoretical analysis. The GSPEN model description in section 3.1 seems rather ad-hoc; no intuition or theoretical insights are given. - Strengths and weaknesses of GSPEN model are not evaluated. Since GSPEN is an extension of SPEN model, it would be great to understand better the settings in which GSPEN is preferred over SPEN and vice-versa.

[Author Response · NeurIPS 2019]

We thank all the reviewers for their valuable feedback. In response we'll include Tab. 1, which gives the average epoch compute time (i.e., compute inference objective and update model weights) during training for our models for each task.

**To Reviewer #2:**

*Re: Fig 1a.* Yes, this should be 'iterations' – we will fix this.

*Re: Variability of different optimization approaches.* It is not the case that choosing an inference method is a "black art." The larger variance of some approaches arises due to an incompatibility of the inference objective and the optimization algorithm used to run inference. For example, unboundedness of the entropy at the boundaries of the domain is known to hurt convergence of Frank-Wolfe for objectives which contain it (see [*]). Furthermore, we present settings where the Struct model was trained with entropy while the $T$ function in GSPEN was trained without entropy or vice-versa. Because the two different model components are trained using different objectives, final performance expectedly suffers. We include those settings for completeness sake, but we do not recommend to use them in practice.

|  | Struct | SPEN | NLStruct | GSPEN |
|---|---|---|---|---|
| OCR (size 1000) | 0.40 s | 0.60 s | 68.56 s | 8.41 s s |
| Tagging | 18.85 s | 30.49 s | 208.96 s | 171.65 s |
| Bibtex | 0.36 s | 11.75 s | – | 13.87 s |
| Bookmarks | 6.05 s | 94.44 s | – | 234.33 s |
| NER | 29.16 s | – | – | 99.83 s |

Table 1: Average time to compute inference objective and complete a weight update for one pass through the training data. We show all models trained for the submission.

**To Reviewer #3:**

*Re: Relaxation of marginal polytope.* The formulation of inference for GSPEN allows the practitioner to select a structured prediction algorithm (and whatever relaxation of $\mathcal{M}$ this entails). Obviously this has consequences for both computational complexity and solution quality. For all of the experiments used in this paper, we use the local marginal polytope $\mathcal{M}_L$ described starting on line 90. We use the structured inference procedure employed in [9, 16, 30]. We think the selected approach provides a good tradeoff between computational complexity and solution quality.

**To Reviewer #4:**

*Re: Theoretical analysis.* We discuss the conditions under which convergence guarantees are available for inference: (1) for Frank-Wolfe starting in line 166 and (2) for structured entropic mirror descent starting in line 188. Unfortunately, convergence guarantees in the settings used for experimentation have not been proved due to non-concavity/non-convexity of the objectives and the employed update steps. However, we think these settings are useful for practitioners.

*Re: Intuition behind GSPEN.* We discuss the motivations for this model starting at line 17: this model permits to use a structured score function (which SPEN does not) while also enabling to use an energy function that scores entire predictions jointly (which is not supported in classic structured prediction).

*Re: SPEN vs. GSPEN.* GSPEN allows a practitioner to augment a structured score function with an additional energy function that jointly scores the prediction vector. Therefore, any setting where structured score functions are used (e.g., for NER – see [1, 29] for examples) can benefit from the GSPEN formulation. Our results demonstrate that for a variety of tasks, having both an energy function and a structured score function results in better performance than having either individually. Specifically, the datasets used for the OCR experiments were designed to demonstrate this: the per-variable information (i.e., predicting from images alone) is noisy, but due to the limited vocabulary used to generate the words, there is much useful information within the structure of the labels that GSPEN is able to exploit. Furthermore, Fig. 3 demonstrates that GSPEN is able to exploit this structure better than either Struct or SPEN. The tradeoff of GSPEN's ability to include structured score functions is its increased computational complexity, which is a consequence of maintaining the structural constraints during inference. Hence, in settings where structure is unknown or strictly higher-order, adding a structured score function to SPEN may not provide benefits performance-wise and will be slower due to the additional cost of structured inference.

*Re: Complexity of inference.* The inference algorithm does not scale based on the number of data points, but rather the number of variables in the problem, the number of regions being modeled in the structured score function, and the number of states each variable takes. The computational complexity of GSPEN depends on the chosen classic structured inference algorithm and its complexity, since it will be called once per iteration of inference. These algorithms scale with the size of the largest region, which is why pairwise structured models are commonly used. The complexity of the structured inference algorithm we use is $O(|\mathcal{R}| \cdot \max_r |P(r)| \cdot \max_r |\mathcal{Y}_r|)$, where $\mathcal{R}$ is the set of graph regions and $P(r)$ is the set of "parents" of region $r$ (in a pairwise model, $P(i)$ is the set of pairs containing variable $i$). Depending on the employed model, computing the gradients of the model may also be costly. For all the problems we consider this cost is lower than the one of computing the structured inference objective.

**References:**

[*] R. G. Krishnan, S. Lacoste-Julien, and D. Sontag. "Barrier Frank-Wolfe for marginal inference." NIPS 2015.


[Meta-Review · NeurIPS 2019]

All the reviewers thought that generalizing the structured prediction energy network (SPEN) to incorporate factored potentials (following graph structure) with proposed approximate inference schemes for structured prediction make a nice contribution to NeurIPS. The extensive experiments were lauded, but concerns were expressed with the theoretical backing of the methods. After discussion and looking at the paper, the AC agrees with R2 that the paper makes an interesting practical contribution, and that the theory could be clarified in follow-up work. The authors should include their timing results as well as additional clarification from the rebuttal in their camera ready version. Additional side notes: - [*] from the rebuttal should be mentioned in the main paper as a way to handle the entropy term over the marginal polytope in a principled manner with Frank-Wolfe. Note that the authors in [*] use line-search for their FW algorithm, which pushes the iterates closer to the boundary (and thus might yield to convergence issues (slow convergence)); I suspect such issues were not observed in this submission as it looks like a fixed step-size scheme was used. - Note that the FW method can fail to converge to an optimum point when the objective is non-differentiable (see example 1 of Nesterov, Math. Prog. 2018, "Complexity bounds for primal-dual methods minimizing the model of objective function", which works for *any step-size* of FW). Given that this submission mentions ReLU activations (which are non-smooth), this caveat should also be mentioned in the paper.